# In situ strain tuning of the metal-insulator-transition of $Ca_2RuO_4$ in angle-resolved photoemission experiments

S. Riccò[1], M. Kim[2,3], A. Tamai [1], S. McKeown Walker[1], F.Y. Bruno [1], I. Cucchi[1], E. Cappelli[1], C. Besnard[1], T.K. Kim [4], P. Dudin [4], M. Hoesch[4,9], M.J. Gutmann[5], A. Georges [1,2,3,6], R.S. Perry[7] & F. Baumberger [1,8]

Pressure plays a key role in the study of quantum materials. Its application in angle resolved photoemission (ARPES) studies, however, has so far been limited. Here, we report the evolution of the $k$-space electronic structure of bulk $Ca_2RuO_4$, lightly doped with Pr, under uniaxial strain. Using ultrathin plate-like crystals, we achieve uniaxial strain levels up to −4.1%, sufficient to suppress the insulating Mott phase and access the previously unexplored electronic structure of the metallic state at low temperature. ARPES experiments performed while tuning the uniaxial strain reveal that metallicity emerges from a marked redistribution of charge within the Ru $t_{2g}$ shell, accompanied by a sudden collapse of the spectral weight in the lower Hubbard band and the emergence of a well-defined Fermi surface which is devoid of pseudogaps. Our results highlight the profound roles of lattice energetics and of the multiorbital nature of $Ca_2RuO_4$ in this archetypal Mott transition and open new perspectives for spectroscopic measurements.

[1] Department of Quantum Matter Physics, University of Geneva, 24 Quai Ernest-Ansermet, 1211 Geneva 4, Switzerland. [2] Centre de Physique Théorique Ecole Polytechnique, CNRS, Universite Paris-Saclay, 91128 Palaiseau, France. [3] College de France, 11 place Marcelin Berthelot, 75005 Paris, France. [4] Diamond Light Source, Harwell Campus, Didcot, UK. [5] ISIS Neutron and Muon Source, Science and Technology Facilities Council, Rutherford Appleton Laboratory, Didcot OX11 0QX, UK. [6] Center for Computational Quantum Physics, Flatiron Institute, 162 5th Avenue, New York, NY 10010, USA. [7] London Centre for Nanotechnology and UCL Centre for Materials Discovery, University College London, London WC1E 6BT, UK. [8] Swiss Light Source, Paul Scherrer Institut, CH-5232 Villigen, PSI, Switzerland. [9] Present address: Deutsches Elektronen-Synchrotron DESY, Photon Science, Hamburg 22607, Germany. Correspondence and requests for materials should be addressed to F.B. (email: Felix.Baumberger@unige.ch)

Mott metal-insulator transitions are driven by electron–electron interactions but often coincide with structural phase transitions[1]. While the latter were long believed to be a secondary response, as argued originally by N.F. Mott[2], realistic numerical studies point to a far more important role of structural changes in stabilizing the Mott state of archetypal insulators[3,4]. This, together with recent theoretical advances, has led to renewed interest in the interplay of lattice energetics and electronic properties near Mott transitions[5–7]. Hydrostatic and uniaxial pressure is particularly important in the experimental study of Mott transitions and also has a profound effect on other emerging properties of quantum materials[1,8–12]. However, conventional pressure cells are fundamentally incompatible with modern surface sensitive spectroscopies, such as angle-resolved photoemission (ARPES). Consequently, the evolution of the $k$-space electronic structure in Mott systems as they are tuned across the metal-insulator transition (MIT) has remained largely unknown. In order to overcome this limitation of ARPES, we developed an apparatus which is compatible with modern ARPES facilities and permits in situ quasi-continuous tuning of uniaxial strain.

Here, we use this capability to investigate the layered perovskite $Ca_2RuO_4$, which is of particular scientific interest as a prototypical multiband Mott insulator. Within band theory $Ca_2RuO_4$ is a good metal with a nearly uniform distribution of the 4 Ru $d$-electrons over the 3 $t_{2g}$ orbitals. How such a multiband metal with fractional occupation can undergo a Mott transition has been debated intensely, but the lack of data from the metallic state has prevented stringent tests of theoretical models[4,13–16]. More recently the magnetic properties in the insulating state of $Ca_2RuO_4$ have attracted much interest[15,17,18] following proposals of a $J_{eff} = 0$ state with excitonic magnetism and an exotic doping evolution[19,20] as well as the observation of unprecedented diamagnetism in a semimetallic phase induced by dc electric current[21].

## Results and Discussion

**Pr doping of $Ca_2RuO_4$.** The insulating state of $Ca_2RuO_4$ is known to be very sensitive to pressure[22–26], chemical substitution[27,28], and even electric fields[21,29]. To further increase the sensitivity of the insulating ground state of $Ca_2RuO_4$ to strain, we have grown a series of La, Nd, and Pr doped single crystals. Details of the sample growth and characterization are given in Supplementary Note 1. Despite the slightly different rare earth ionic radii these samples behave qualitatively similarly. We thus chose to concentrate on $Ca_{2-x}Pr_xRuO_4$ with $x = 0$, 0.03, 0.04, and 0.07. The main effect of doping is to suppress the structural phase transition accompanying the metal-insulator transition (MIT) from $T_{MI} \sim 360$ K for $x = 0$ to $\sim 85$ K at the highest Pr doping level of $x = 0.07$ used in our study (see phase diagram in Fig. 1a).

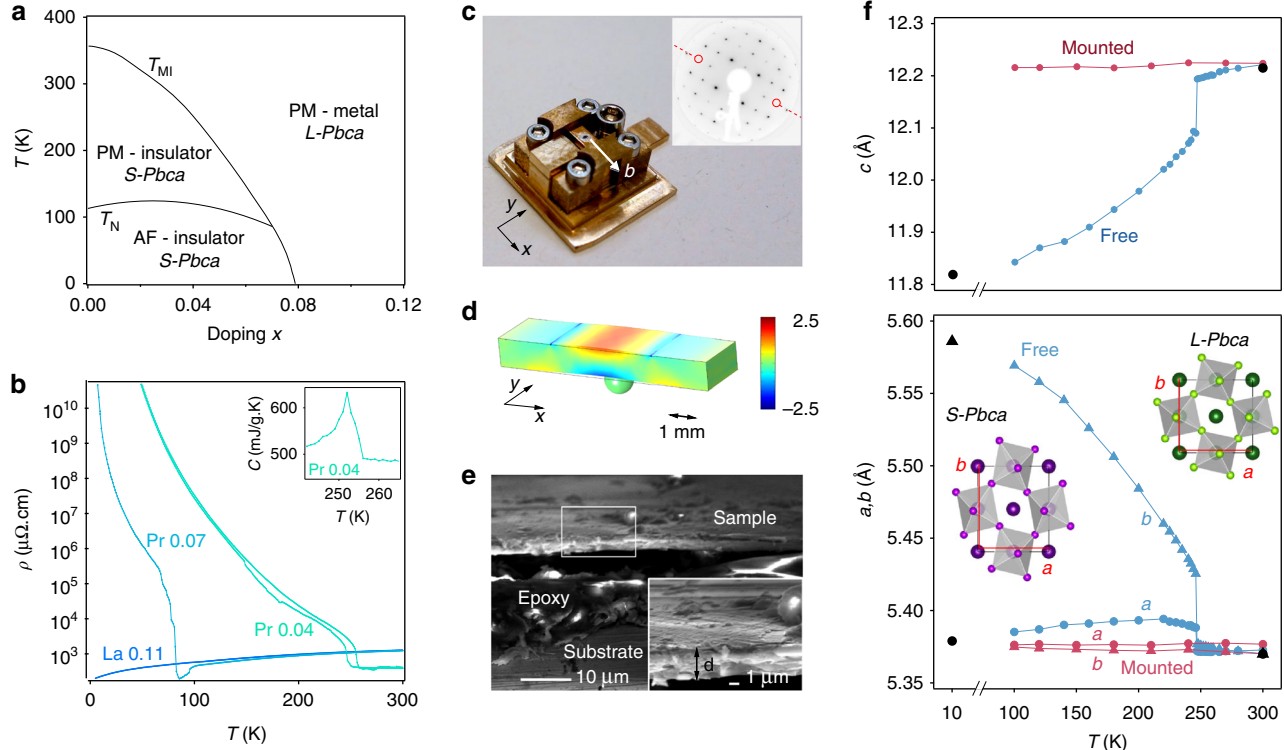

**Fig. 1** Phase diagram and strain apparatus. **a** Metallic and insulating phases of $Ca_{2-x}Pr_xRuO_4$ are separated by a first order structural phase transition from $L$-Pbca to $S$-Pbca. Canted antiferromagnetism is observed in all insulating samples below ≈110 K. **b** Resistivity curves for $Ca_{2-x}Pr_xRuO_4$ ($x = 0.04$, 0.07) and $Ca_{2-x}La_xRuO_4$ ($x = 0.11$), which we use as a reference for the metallic ground state. The hysteretic behavior shown for $x = 0.04$ confirms that the transition is first order. The inset shows the peak in the specific heat at the structural phase transition for $x = 0.04$. **c** Photograph of the strain apparatus. Bending the substrate along the $b$-axis drives strained $Ca_{2-x}Pr_xRuO_4$ toward the insulating orthorhombic $S$-Pbca ground state. The inset shows a LEED pattern of a strained sample, revealing the glide plane (dashed red line). **d** Calibration of the strain apparatus using finite element analysis. The color scale encodes the tensile strain $\varepsilon_{xx}^{bend.}$. **e** Scanning electron micrograph of a cleaved and fully strained sample. The black region between sample and epoxy layer is due to a shadowing effect caused by the high roughness of the cut through sample and substrate. The inset shows a higher magnification image of the area indicated by a white rectangle. **f** Temperature dependence of the lattice constants for $x = 0.04$ measured by single crystal X-ray diffraction (XRD) before and after mounting the sample on our strain apparatus. We find that samples as thin as the one imaged in **e** preserve the high-temperature $L$-Pbca structure down to base temperature. Black symbols indicate lattice constants obtained by single crystal neutron diffraction at 10 K and 300 K

Our single crystal neutron diffraction data show that in line with the suppression of $T_{MI}$, the ground state crystal structure of $Ca_{2-x}Pr_xRuO_4$ undistorts progressively towards the metallic high-temperature state of pure $Ca_2RuO_4$ as the doping is increased, which facilitates strain tuning of the MIT (Supplementary Figure 1). Nevertheless, the structural transition of $Ca_{2-x}Pr_xRuO_4$ retains the characteristics of the phase transition in pure $Ca_2RuO_4$. In particular, it is symmetry-preserving for all doping levels and mainly characterized by a substantial flattening of the $RuO_6$ octahedra together with an elongation of the $b$-axis leading to strong orthorhombicity in the insulating phase [Supplementary Figure 1]. We will exploit this latter property to tune the MIT by uniaxial strain. Adopting the notation used for pure $Ca_2RuO_4$, we call the metallic phase with long $c$-axis and $Pbca$ space group $L$-$Pbca$ and the insulating phase with short $c$-axis $S$-$Pbca$.

Importantly, Pr doping does not introduce itinerant carriers in the $S$-$Pbca$ phase of $Ca_{2-x}Pr_xRuO_4$. This is evident from the highly insulating nature of our $Ca_{2-x}Pr_xRuO_4$ samples, which is fully consistent with an earlier study on La-doped $Ca_2RuO_4$[28]. Compared to lightly doped cuprates or iridates, $Ca_{2-x}Pr_xRuO_4$ is several orders of magnitude less conductive (see Supplementary Figure 2), which implies a complete localization of the extra electron supplied by the Pr ion. Such a localization of doped carriers can arise from polaronic effects or from a Mott transition in the impurity band[30], and is not uncommon in chemically doped Mott insulators[1,31,32].

**Strain tuning apparatus.** Our in situ transferable strain apparatus is shown in Fig. 1c. It is actuated mechanically by turning a screw, which causes a lever to press a stainless steel ball from below on a 1 mm thick CuBe substrate. The elastic deformation of the substrate results in tensile strain $\varepsilon_{xx}^{bend.}$ along the bending direction on the upper surface and a much smaller compressive strain $\varepsilon_{yy}^{bend.}$ in the orthogonal direction. We calibrate $\varepsilon^{bend.}$ using finite element analysis, as shown in Fig. 1d and Supplementary Figure 3, taking into account the indent in the substrate left by the ball, which we measure at the end of each experiment. For a maximal coupling of in-plane strain to the $c$-axis compression, which putatively drives the MIT[4], we align the crystalline $b$-axis with the bending direction. As this axis lies in a glide plane of the $Pbca$ structure, it can be identified readily in low-energy electron diffraction (LEED) patterns via the extinction of spots at certain energies (inset to Fig. 1c).

Key to our experiment is the exploitation of the initial compressive strain $\varepsilon^i$ exerted by the large differential thermal contraction as apparatus and sample are cooled to base temperature. The strain $\varepsilon^i$ is dominated by the negative thermal expansion of $Ca_{2-x}Pr_xRuO_4$ along the $b$-axis and is thus highly uniaxial [see Fig. 1f and Supplementary Note 3]. Using literature data for the CuBe substrates and our neutron diffraction data for $Ca_{2-x}Pr_xRuO_4$, we calculate nominal values of $\varepsilon_{xx}^i = -4.1\%$ ($-2.3\%$) for Pr concentrations $x = 0.04$ (0.07) and approximately an order of magnitude lower values for $\varepsilon_{yy}^i$. We directly confirm these exceptionally high strain levels for the most challenging case of a $x = 0.04$ sample using X-ray diffraction on cleaved samples mounted on our strain apparatus. From the data shown in Fig. 1f we calculate an initial compressive strain $\varepsilon_{xx}^i = (b^{mounted} - b^{free})/b^{free} = -3.6\%$ at 100 K, in excellent agreement with the nominal value of $-3.8\%$ at this temperature. These strain levels are achieved by mounting ultrathin plate-like single crystals to minimize strain relaxation. Cross-sectional electron microscopy images of our mounted and cleaved samples indicate typical thicknesses of ~10 μm for the epoxy layer and 2–10 μm for the single crystals (Fig. 1e). Having confirmed negligible relaxation at the highest strain used in our experiment, we approximate the total strain as $\varepsilon^{tot} = \varepsilon^i + \varepsilon^{bend.}$, where $\varepsilon^i$ is compressive and $\varepsilon^{bend.}$ tensile.

**Strain-induced metallic state.** The striking effect of uniaxial strain on the electronic structure of $Ca_{2-x}Pr_xRuO_4$ is evident from the ARPES Fermi surfaces shown in Fig. 2a, b. For an unstrained sample in the $S$-$Pbca$ phase we find negligible intensity at the Fermi level $E_F$ and no discernible structure in momentum space consistent with a gapped Mott insulating state. In a fully strained sample with $L$-$Pbca$ structure, on the other hand, a clear Fermi surface emerges, demonstrating a metallic ground state. Intriguingly, the strain-induced metallic state differs strongly from lightly doped cuprates and iridates[33,34]. In particular, we find no anisotropy in the quasiparticle coherence and no evidence for a pseudogap along the entire Fermi surface within the precision of our experiment of $\approx 2$ meV.

The Fermi surface of strained $Ca_{2-x}Pr_xRuO_4$ is remarkably simple considering the large unit cell containing 4 formula units and 16 electrons in the Ru $t_{2g}$ shell. We find a square hole-like sheet centered at $\Gamma$, which encloses a smaller electron-like Fermi surface, and four small lens-shaped sheets at the X and Y points, respectively, as summarized in Fig. 2d. The absence of exchange splitting in our experimental data indicates a paramagnetic metallic state, as it is also observed in the high-temperature $L$-$Pbca$ phase of undoped $Ca_2RuO_4$ and for the ground state of

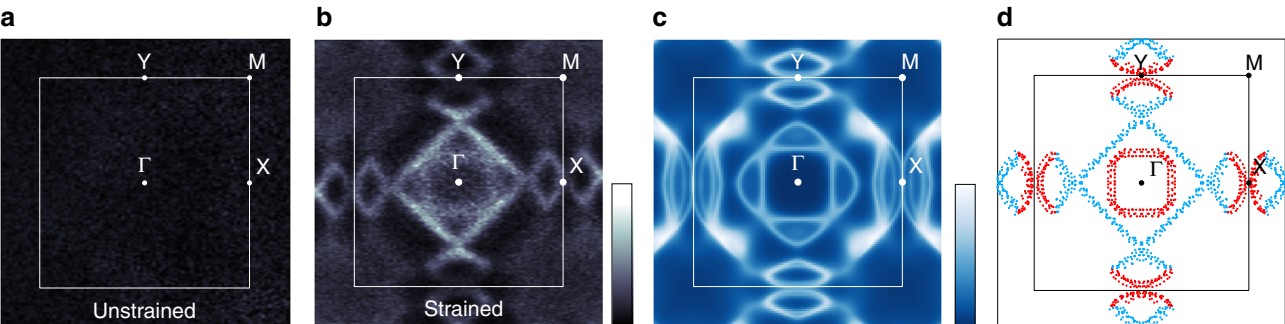

**Fig. 2** Strain-induced metallic state of $Ca_{2-x}Pr_xRuO_4$. **a, b** ARPES Fermi surface maps for a $Ca_{2-x}Pr_xRuO_4$ sample with $x = 0.07$ and a fully strained $x = 0.04$ sample measured at 50 K and 8 K, respectively. The former was measured on a sufficiently thick sample to cause almost complete relaxation of the initial strain. The data were acquired using 64 eV photons with linear horizontal polarization. Light colors correspond to high intensities. **c** Dynamical mean field theory (DMFT) calculation of the Fermi surface. For details, see methods. Light colors correspond to high intensities. **d** Fermi surface contours extracted from the data in **b**. Contours originating predominantly from the quasi-1D $xz/yz$ orbitals and the in-plane $xy$ orbital are colored in light blue and red, respectively

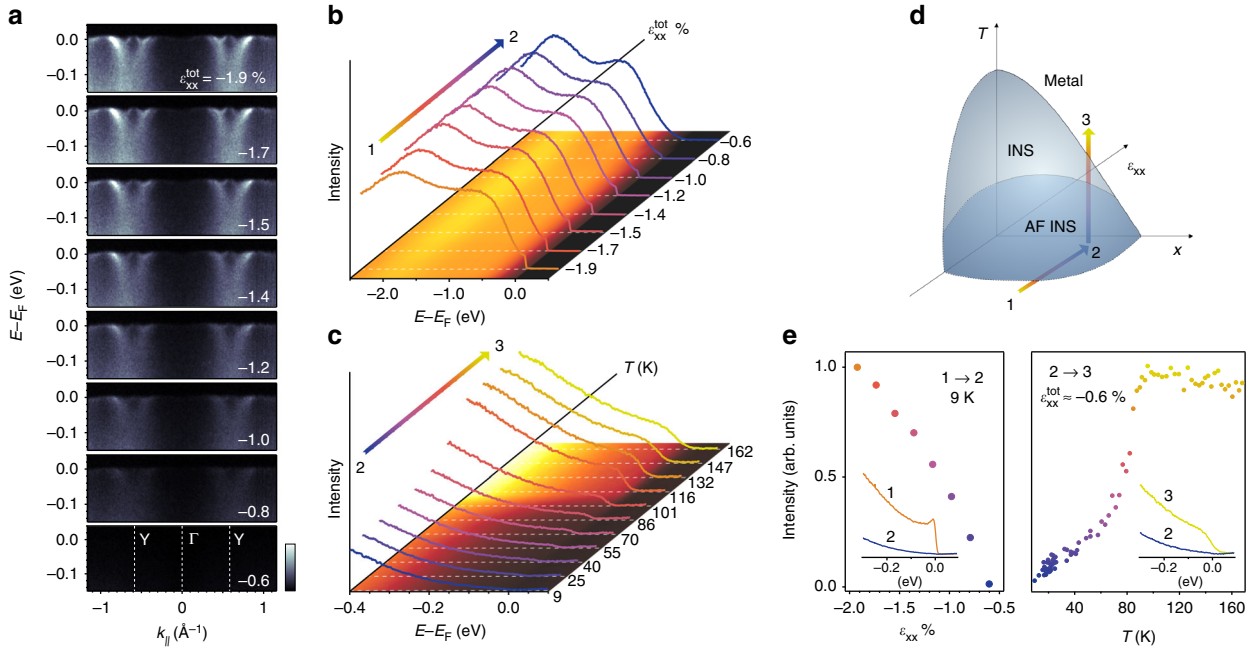

**Fig. 3** Strain tuning of the MIT. **a** Evolution of the quasiparticle band structure of $Ca_{2-x}Pr_xRuO_4$ for $x = 0.07$ along YΓY at 8 K as the strain is tuned along the path $1 \rightarrow 2$ in the schematic phase diagram shown in **d**. The data were acquired using 64 eV photons with linear horizontal polarization. Light colors correspond to high intensities. **b** Angle-integrated energy distribution curves (EDCs) over the full width of the occupied Ru $t_{2g}$ states as a function of uniaxial strain. **c** Angle-integrated EDCs as a function of temperature measured at minimum strain $\varepsilon_{xx}^{tot} \approx -0.6\%$ (path $2 \rightarrow 3$). The sample undergoes the MIT at ~90 K. **e** Evolution of the spectral weight at the Fermi level along the path $1 \rightarrow 2 \rightarrow 3$ defined in **d**

highly La-doped $Ca_2RuO_4$ with $L$-$Pbca$ structure[28]. Remarkably though, a paramagnetic low-temperature phase differs from pure $Ca_2RuO_4$, where itinerant ferromagnetism with an ordered moment of 0.1 ÷ 0.3 $\mu_B$ is observed below ~20 K under hydrostatic pressure[22], uniaxial stress[35] and in epitaxially strained thin films[25,26]. Our results thus provide further evidence for a generic proximity of metallic ruthenates to magnetically ordered states induced by an instability of the Fermi surface[36,37].

Earlier dynamical mean field theory (DMFT) calculations of the MIT in $Ca_2RuO_4$ predict that the full $d_{xy}$ orbital polarization with $n_{xy} \approx 2$, $n_{xz} \approx n_{yz} \approx 1$ and $p = n_{xy} - (n_{xz} + n_{yz})/2 \approx 1$ observed in the insulating state vanishes in the metallic $L$-$Pbca$ phase[4,7,14,16,38,39]. This is qualitatively consistent with our experiments showing signatures of all three $t_{2g}$ orbitals on the Fermi surface. Identifying the extended straight sections (light blue in Fig. 2d) with the quasi-1d $xz$, $yz$ orbitals and the curved sections of the lens-pockets (red) as well as the circular pocket at Γ with $xy$ character, we estimate $n_{xy} \approx 1.2$ from a simple tight-binding model. This is in fair agreement with the theoretical prediction of 4/3 and substantially reduced from the value of $n_{xy} \approx 2$ of the insulating state[4,14]. However, the validity of such a model is questionable. In the presence of spin-orbit coupling (SOC), individual Fermi surface sheets can no longer be identified with a single orbital character. We thus performed DMFT calculations of the Fermi surface including SOC following the method described in ref. [40]. As shown in Fig. 2c, these calculations reproduce the two sheets centered at Γ as well as the lens-like pockets on either side of the X and Y points. The only significant discrepancies are a small splitting around X, which is not resolved experimentally, and an intense feature on the Brillouin zone diagonal arising from a band that is unoccupied in experiment but touches the chemical potential in our calculation. Differences on such a small energy scale are well within the precision of our computational approach and do not compromise the overall excellent agreement with experiment.

This constrains the orbital polarization of the metallic state to within a few percent of the value $p = -0.08$ obtained in our calculation and thus provides compelling evidence for a collapse of the polarization at the MIT.

**Strain-tuning of the MIT.** In Fig. 3, we demonstrate in situ tuning across the Mott transition using our strain apparatus. We first focus on the evolution of the near-$E_F$ electronic structure for a $x = 0.07$ sample following the path $1 \rightarrow 2$ in the schematic phase diagram of Fig. 3d. A cut along the ΓY high-symmetry line in a fully strained sample (Fig. 3a, first panel) shows well-defined, strongly renormalized quasiparticle states at very low energy only indicating a delicate Fermi liquid regime. Beyond a coherence scale of ~30 meV, the excitations broaden rapidly and their dispersion increases simultaneously. These high-energy states can be tracked down to ~ −2.7 eV and thus essentially over the full bare bandwidth (Fig. 4). Such a coexistence of heavy quasiparticles with unrenormalized high-energy states was identified as a hallmark of Hund's metals with profound implications on magnetic susceptibility, thermal and electrical transport[41]. Reducing the uniaxial strain by bending the substrate, we fully recover the characteristic spectrum of insulating $Ca_{2-x}Pr_xRuO_4$ with an exponential onset of weight. Interestingly though, the quasiparticle dispersion is not affected strongly by strain. We can thus exclude that the strain-induced Mott transition is triggered by a divergence of the effective mass predicted in the Brinkmann-Rice model[42]. Raising the temperature ($2 \rightarrow 3$, Fig. 3c), the insulating state undergoes another phase transition close to $T_{MI}$ of the unstrained state and we recover a metallic spectrum with significant weight at $E_F$. As shown in Fig. 3e, the suppression of the spectral weight at $E_F$ during the strain-tuning is gradual. This can either indicate a second order phase transition or a phase coexistence with domains below the lateral dimension of ≈20 × 50 μm probed by ARPES. Given the sensitivity of the electronic state of

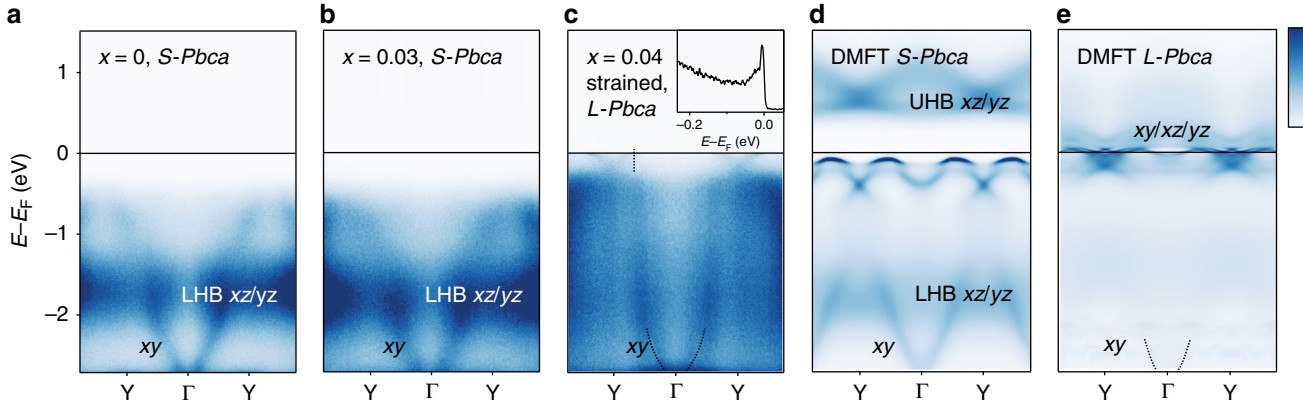

**Fig. 4** Redistribution of spectral weight across the MIT. ARPES spectral weight along $\Gamma Y$ measured for different dopings and structures: **a** undoped $Ca_2RuO_4$ at 180 K (S-Pbca, paramagnetic); **b** $Ca_{2-x}Pr_xRuO_4$ with $x = 0.03$ at 150 K (S-Pbca, paramagnetic); **c** fully strained $Ca_{2-x}Pr_xRuO_4$ ($x = 0.04$) at 10 K (L-Pbca, metallic). We display a superposition of data acquired with left and right circular polarization; the inset in **c** shows a clear quasiparticle peak at $k = k_F$ (dotted line); **d**, **e** DMFT calculation for the S-Pbca and L-Pbca structure along the same cut. For details, see methods. The bottom of the $xy$ band, which becomes visible upon enhancing the contrast in the DMFT calculation of the metallic state is indicated by a dotted line in **c**, **e**. Dark colors correspond to high intensities

$Ca_{2-x}Pr_xRuO_4$ to its first order structural phase transition, we consider the latter more likely. Additional evidence for phase coexistence, which was also observed in diffraction experiments on $Ca_2RuO_4$ under hydrostatic pressure[23], is shown in Supplementary Figure 4.

**Redistribution of spectral weight across the MIT.** ARPES data taken over a larger energy range show a remarkable redistribution of spectral weight at the strain-induced MIT. In Fig. 4a–c, we compare dispersion plots covering the full width of the $t_{2g}$ shell from two insulating and paramagnetic samples with S-Pbca structure and $x = 0$, 0.03 with a fully strained sample of comparable doping $x = 0.04$ in the L-Pbca structure. The first important conclusion from this data is that light Pr doping alone causes minor changes in the electronic structure only. Its main effect is a small shift of the chemical potential. Importantly though, $E_F$ remains in the correlated gap. This strongly supports the notion of fully localized dopants in the insulating S-Pbca phase inferred previously from the highly insulating nature of these samples. The main spectroscopic features in the insulating phase are a dispersive state with $\approx 2$ eV bandwidth and an intense non-dispersive peak at $-1.7$ eV. With reference to our dynamical mean field theory (DMFT) calculations (Fig. 3d) and consistent with ref. [16], we identify these features with the fully occupied $xy$ orbital and the lower Hubbard band (LHB) of $xz/yz$ character, respectively. This confirms a basic electronic configuration in the insulating S-Pbca phase with fully occupied $d_{xy}$ orbital and half-filled $d_{xz}/d_{yz}$ bands split into lower and upper Hubbard band, as proposed on the basis of DMFT calculations[4,14].

Straining a lightly doped $Ca_{2-x}Pr_xRuO_4$ sample, we observe a substantial redistribution of spectral weight. Most notably, the intensity in the LHB collapses suddenly across the MIT and coherent quasiparticle states appear at the chemical potential (Fig. 3c). Both of these effects are reproduced by our DMFT calculations shown in Fig. 3e. Interestingly though, the sudden collapse of the LHB is in stark contrast to lightly doped cuprates, where metallicity emerges from a gradual transfer of spectral weight from the LHB to coherent quasiparticle states[43]. We interpret this as a generic manifestation of the additional orbital degree of freedom of multiband Mott insulators. In effective single band systems, such as the cuprates or iridates, the orbital occupancy can only change by the small number of doped carriers, resulting in dominantly half-filled sites retaining a strong

memory of the Mott phase. In $Ca_{2-x}Pr_xRuO_4$, on the other hand, our data show a discontinuous change of the $n_{xz/yz}$ occupancy from 1 to $\approx 4/3$ in spite of the only light doping because of substantial interorbital charge transfer across the MIT. The large deviation from half filling causes a sudden collapse of the LHB and renders electronic energies comparable to lattice energies under strain, resulting in a strongly first order nature of the Mott transition.

**Conclusions.** Our results show that tuning uniaxial strain in ARPES experiments is a promising method to study phase transitions or, more generally, structure—property relations of quantum materials. Potential applications of our method range from tuning magnetism, to topological phase transitions, two-dimensional van der Waals materials and unconventional superconductors showing large responses to strain, such as $Sr_2RuO_4$[11].

## Methods

**Sample preparation and ARPES experiments.** Single crystals of $Ca_{2-x}Pr_xRuO_4$ were grown through the floating zone (FZ) technique using a Crystal System Corporation FZ-T-10000-H-VI-VPO-I-HR-PC four mirror optical furnace. Samples were grown in 90% oxygen pressure, and the initial Ru concentration in the polycrystalline rods was about 20% higher than the nominal value to compensate for evaporation during the growth. The bulk properties were thoroughly characterized by resistivity, specific heat, magnetization measurements, and single crystal neutron diffraction at the ISIS spallation neutron source[44]. Doping levels were measured by energy and wavelength dispersive X-ray spectroscopy (EDX/WDX) and were found to be systematically lower by 20–30% than in the polycrystalline growth rod. Angle-resolved photoemission spectroscopy (ARPES) experiments were performed at the I05 Beamline of the Diamond Light Source[45]. The presented data were acquired with linearly and circularly polarized light at 64 eV photon energy and an overall resolution of $\approx 12$ meV/0.015 Å$^{-1}$.

**Single crystal neutron diffraction.** The data were acquired at the single crystal diffractometer SXD of the ISIS spallation neutron source using the time-of-flight Laue technique[44]. We measured undoped $Ca_2RuO_4$, as well as La, Pr and Nd doped single crystals at 10 K and 300 K. For La doping $x = 0.11$ data were only acquired at 4 K because of the complete suppression of the structural phase transition. The crystal structures together with further details on data collection and refinement are reported in the CIF files available online and summarized in Supplementary Tables 1–3.

**DMFT calculations.** Electronic structure calculations within DFT + DMFT were performed using the full potential implementation in the TRIQS library[46]. The density functional theory (DFT) part of the computations used the local density approximation implemented in the Wien2k package[47]. Wannier-like $t_{2g}$ orbitals were constructed out of Kohn-Sham bands within the energy window $[-2, 1]$ eV with respect to the Fermi energy. We used the full rotationally invariant Kanamori

interaction with $U = 2.3$ eV and $J = 0.4$ eV, which successfully describes correlated phenomena of other ruthenates[40,41]. The quantum impurity problem was solved using the continuous-time hybridization-expansion Monte Carlo impurity solver as implemented in the TRIQS library. The calculation of the insulating state shown in Fig. 4d of the main text was done for the experimental crystal structure of undoped $Ca_2RuO_4$ at 10 K described in Supplementary Note 1 and did not include SOC. Calculations of the metallic L-Pbca phase (Figs. 2c, 3e) included SOC following the methods described in ref. [40]. We verified that the orbitally diagonal elements of the self-energy matrix are not affected by SOC and that the off-diagonal elements are frequency independent up to high energies and can thus be treated as a correlation enhanced SOC. Due to the lack of structural data from strained $Ca_{2-x}Pr_xRuO_4$, we used the ground state L-Pbca crystal structure of $Ca_{1.89}La_{0.11}RuO_4$ reported in Supplementary Table 2. The doping of $x = 0.11$ was included in the DMFT part of this calculation, while the DFT step was done for stoichiometric $Ca_2RuO_4$. Reducing the doping to $x = 0.04$ in the metallic phase has a minor effect on the DMFT self-energies only. The Fermi surface shown in Fig. 2c was computed with the orbitally diagonal self-energy from a DMFT computation without spin-orbit coupling and the frequency independent orbitally off-diagonal elements from the calculation with SOC.

## Data availability

The data relevant to the findings of this study are available from the corresponding authors on reasonable request.

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

## Acknowledgements

We thank E. Giannini, D. McMorrow, D. Pincini, D. Jaccard, C. Renner, M. Spera, V. Pasquier for discussions, J. Teyssier for Raman experiments on $Ca_{2-x}Pr_xRuO_4$, M. Spera for help with the numerical strain simulations and R. Pellet for machining and aligning the strain apparatus. The experimental work was supported by the Swiss National Science Foundation (200021-153405 and 200020-165791). Theoretical work was supported by the European Research Council grant ERC-319286-QMAC, the NCCR MARVEL of the SNSF and the Simons Foundation (Flatiron Institute). Crystal growth and characterization at UCL was supported by the EPSRC grant EP/N034694/1. We gratefully acknowledge the Science and Technology Facilities Council (STFC) for access to neutron beamtime at ISIS, and also for the support of sample preparation at the UCL crystal growth laboratory. We would like to thank G. Stenning for help on the Smartlab XRD and Quantum Design MPMS instruments in the Materials Characterisation Laboratory at the ISIS Neutron and Muon Source. We acknowledge Diamond Light Source for time on beamline I05 under proposal SI17381.

## Author contributions

S.R. and R.S.P. grew $Ca_{2-x}Pr_xRuO_4$ single crystals. S.R., A.T. S.M.W., F.Y.B., I.C., E.C., T.K.K., P.D., M.H. and F.B. prepared and carried out the ARPES experiments. T.K.K, P.D., and M.H. were responsible for the synchrotron beamline. A.G. and M.K. performed the DMFT calculations. M.J.G. performed the neutron diffraction experiments and data refinement. C.B. carried out the XRD experiments. A.G. directed the theoretical work. A.T. and F.B. planned the research and directed the experimental work. All authors contributed to the manuscript.

## Additional information

**Competing interests:** The authors declare no competing interests.

