## [Peer Review File · Nature Communications]

Reviewers' comments:

Reviewer #1 (Remarks to the Author):

The authors used specially designed device to perform strain dependent ARPES experiments on lightly Pr doped Ca₂RuO₄. They were able to switch between metallic and insulating states in situ and obtain electronic structures of metallic Ca₂RuO₄ which has not been reported before. This work is, to my knowledge, the first report on in situ uniaxial strain dependent ARPES experiments. In addition, they could address the MIT mechanism in Ca₂RuO₄ in electronic structure point of view. These are important achievement and therefore I would in principle recommend the manuscript for publication in Nat Comm, but I would like the authors to answer one major comment below.

1. Their interpretation of the MIT is interesting. On the other hand, the discussion regarding the abrupt occupation change may need further proof. In the 1st paragraph in page 8, the authors state that "Due to the large rotation of the RuO₆ octahedra and the sizeable spin-orbit coupling, we cannot uniquely identify the orbital character on the Fermi surface from linear dichroism measurements", describing the issues on assigning a specific orbital to a band. However, they later discuss orbital occupancies even though an orbital cannot be assigned to a band. Obviously, these two do not go together. A more direct way to measure the occupation should be polarization dependent X-ray absorption. Can the authors compare their interpretation with X-ray data? In addition, what does DMFT result say on the occupation?

Here are relatively minor comments for the authors to consider.

1. My understanding is that they used Pr doped system which is closer to the MIT and therefore easier to move across the phase boundary by strain. If it is the case, it needs to be explained in the introduction.
2. In the 2nd paragraph, the authors state "Consistent with a previous study on La-doped Ca₂RuO₄ [28]...." In that statement, they claim that La and Pr do not introduce itinerant carriers. However, A site rare-earth atom substitution introduces carriers in similar systems such as (La-Sr)₂RuO₄. In addition, in the work in reference 28 only asserts that the doped electrons do not contribute to the magnetism but does not rule out possibility for carrier doping. So, either the claim should be backed by evidences or the statement needs to be revised?
3. The authors say "However, it appears plausible to interpret the extended straight sections of the experimental Fermi surface as originating predominantly from the quasi-1d xz,yz orbitals while the curved sections of the lens-pockets as well as the circular pocket at Gamma are likely of dominant xy character." It may not be easy for the readers to understand all these if they are not familiar with the electronic structures of ruthenates. For example, it would be easier to understand if lens-pocket and quasi-1d band are marked and described.
4. It is somewhat peculiar to see the Fermi surface in Fig 2b FS indicates very small hybridization between 'yz' and 'xz' bands in spite of the rotation. Is it consistent with DMFT? Can surface be different from bulk?
5. Technical question 1: Not all the strain in CuBe shown in Fig 1d may be transferred to the sample because epoxy can have elasticity? XRD has been measured to obtain the true strain but one should not assume the strains on the surfaces of CuBe and sample are the same.
6. Is sample bent at all? Is bending truly negligible?

Reviewer #2 (Remarks to the Author):

This paper mainly reports the evolution of the electronic states across the MIT in Ca_{2-x}Pr_xRuO₄ based on ARPES measurements. There are three main tuning parameters used in this study: {1} Light electron doping with Pr³⁺ in place of Ca²⁺; {2} (nearly) biaxial substrate-strain with the BeCu substrate; and {3} (additional) uniaxial bending-strain with a new device. The effects of

these three parameters are presented in a mixed and confusing way, as I describe below. Especially one of the main conclusions that the observed metallic state using $\{1\}$, $\{2\}$ and $\{3\}$ "represents the intrinsic metallic phase of Ca_2RuO_4 " (the last paragraph on page 7) is questionable. Thus, although the paper contains interesting new results, I do not recommend it for publication in Nature Communication, unless major revision is made to the manuscript. Below, I list specific comments for revision.

(1) In the title, " Ca_2RuO_4 " should be changed to " $\text{Ca}_{2-x}\text{Pr}_x\text{RuO}_4$ ", since the main results in this paper is not based on "pure Ca_2RuO_4 ".

(2) In the abstract, "previously unexplored metallic state at low temperatures" should be rephrased to "previously unexplored electronic structures of the metallic state at low temperatures", since there are already a few reports on pressure-induced metallic state, ref (23) and Alireza et al., J. Phys.: Condens. Matter 22 (2010) 052202. The latter reported superconductivity as well.

(3) The abstract should clearly state the roles of $\{1\}$, $\{2\}$ and $\{3\}$ above. The present abstract states as if "uniaxial strain" (the first sentence) of -4.1% has been achieved. In my understanding, -4.1% refers to biaxial substrate-strain and also in the ab-plane (not mentioned).

(4) In captions of Fig. 1, again the roles of $\{1\}$, $\{2\}$ and $\{3\}$ are mixed and confusing. Figs. a and b represent $\{1\}$. Figs. e and f represent a "standard substrate effect" due to $\{2\}$ without "bending". Figs. c and d are related to "additional" strain by $\{3\}$. Thus, it is natural to align (a,b)(f,e)(c,d) in this order.

The explanation of the inset of Fig. e is needed in the caption.

The notation ϵ^{sub} in Fig. d is confusing. See (5) below.

(5) On page 5, the major strain introduced in this study, ϵ^{i} , is biaxial substrate-strain due to differential thermal contraction between the sample and the BeCu substrate. According to Fig. 1 f, the a-axis is also strongly compressed. Thus I suggest using $\epsilon^{\text{total}} = \epsilon^{\text{substrate}} + \epsilon^{\text{bending}}$ or $\epsilon^{\text{total}} = \epsilon^{\text{biaxial}} + \epsilon^{\text{uniaxial}}$ to clarify the notations. By the way is it correct to write $\epsilon^{\text{i}}_{\text{xx}} = \epsilon^{\text{i}}_{\text{yy}} = -4.1\%$ here? If the assessment of $\epsilon^{\text{i}}_{\text{yy}}$ is difficult, the author should write so in the main text; I notice no description in the Supplement, either.

(6) Concerning the first line on page 5, a very relevant literature explaining the anisotropic effects of uniaxial in-plane strains on T_{N} is Taniguchi et al., Phys. Rev. B 88, 205111 (2013), "Anisotropic uniaxial pressure response of the Mott insulator Ca_2RuO_4 ".

(7) The last paragraph on page 7: Persuasive argument is need to explain why the doped and uniaxially strained system "represents the intrinsic metallic phase of Ca_2RuO_4 ", rather than hydrostatically compressed pure Ca_2RuO_4 with the ferromagnetic metallic state and superconductivity.

(8) The first paragraph on page 8: The discussion and conclusion in this paper heavily rely on the assignment that "the curved sections of the lens-pocket ... are likely of dominant xy character". If the authors mean that the other half (straight sections) of the lens-pocket is of yz/zx character, they should write it clearly. Using two different colors in the extracted Fig. 2 c may help.

(9) In the third paragraph on page 8: "The first important conclusion from this data is that light Pr-doping alone causes minor changes in the electronic structure only." This is clearly an under-evaluation of the doping effect, since ferromagnetic ordering disappears in the metallic state under strains.

(10) Comments to improve the figures and captions:

10-1. The compound name $\text{Ca}_{2-x}\text{Pr}_x\text{RuO}_4$ should be clearly written in the captions of Figs. 2 a,b, and Fig. 3.

10-2. Some symbols are not visible due to poor color combinations: The use of red color in the dark background in Figs. 1 a and inset of e, and the use of black "2" in the dark blue background in Fig. 2 d.

10-3. Fig. 2 d: Is "epsilon" here ϵ^{total} ? "%" should be deleted. Since warming severely reduces the magnitude of ϵ^i , the straight line 2-3 should be curved towards positive epsilon on increasing T. More importantly, do the author anticipate paramagnetic insulator state persisting to $T=0$ as Fig. 2 d indicates? Is it based on the $J_{\text{eff}}=0$ model [18-21]? If so, are the DMFT results in Fig. 3 d and e consistent with the $J_{\text{eff}}=0$ model (in terms of spin-orbit interaction and crystal-field splitting within t_{2g}) ?

(11) The roles of {2} and {3} are not clearly stated and sometimes misleadingly used.

11-1. Fig. 2 g caption: $\epsilon^i \sim -0.6\%$ here may be ϵ^{total} .

11-2. Second paragraph on page 9: "Straining" here means biaxial substrate-strain since Fig. 3 are all about {2} WITHOUT {3}.

(12) In Fig. 1 b for $x(\text{Pr}) = 0.07$, why does ρ first drops just below 100 K?

Reviewer #3 (Remarks to the Author):

The manuscript of Ricco et al. reports angle-resolved photoemission (ARPES) experiments for the strained Ca_2RuO_4 compound, driving it from the conventional Mott-insulating state at low temperature to a seemingly Fermi liquid by elongation along the c-axis.

The text is well written and provides a proper description of the system and the findings along with a careful data analysis. Additional theoretical calculations support the experimental results. Overall this is a nice work and paper, especially highlighting the novel possibility of conducting ARPES measurements with mechanically applied strain (contrary to usual epitaxial strain) to the sample. This opens the door to various applications to other demanding materials cases.

But one main problem of the work remains. The findings, results and outcome in terms of physics are not that surprising, in fact, they more or less reproduce the picture of Ca_2RuO_4 that has been established/expected from theory (e.g. Ref. 4) and other related experiments (e.g. Ref. 17). For instance, the strong change of the orbital polarization from insulator to metal was noted in Ref. 4. Furthermore, the metallic state that is found in here is seemingly 'simple' without true unexpected features. Moreover, the theory comparison is somewhat performed on the 'easy' side, namely by reproducing the known bulk spectra of S-Pbca and L-Pbca. No attempt was made to bring in the actual doping or strain of the present experimental scenario, e.g. to study how sudden the orbital-polarization change truly is.

This is a powerful and strong manuscript in terms of experimental capabilities, but with a rather limited aspect of novelty in terms of ruthenate- or general Mott physics. Put differently, the single fact that one can drive a Mott insulator into a metal is per se not that surprising, everybody working in this field of physics is aware of it. Surely here, the experimental techniques behind this are of major quality.

The work may be suitable for Nature Communications because of its strong technical aspects, but it would be even more so, if some new physics could be drawn from this study.

We thank the referees for their careful work in assessing our manuscript. Below we repeat the comments of the referees in italics and provide a point-by-point response. Text in black italics is paraphrased, while all the blue text is copied from the reports.

Reviewer 1

The main comment of reviewer 1 concerns the precise orbital occupation. In particular, he/she writes “the abrupt occupation change [across the MIT] may need further proof” and asks about X-ray absorption (XAS) data and the result of DMFT on the occupation.

Motivated by these comments and related concerns of the other referees, we have included additional DMFT calculations in the manuscript. In addition, some of our collaborators performed XAS measurements on the same samples. However, we prefer not to include these measurements since their interpretation is not fully understood. Most importantly, and fully consistent with the literature [Mizokawa et al. PRL 87, 077202 (2001)] we find that the standard single-ion interpretation of our XAS data does not reproduce the well-established 2-1-1 orbital occupation (full xy orbital) of the insulating state of pure Ca₂RuO₄. Taken at face value, XAS obtains between 0.2 and 0.5 holes in the xy-sheet. However, it is near impossible to obtain a Mott-insulating state for such a fractional occupation. This strongly suggests a problem with the interpretation of XAS. In part, the failure of XAS to reproduce the established picture of the insulating phase of Ca₂RuO₄ might be attributed to the peculiar coexistence of highly delocalized band-insulating xy states and fully localized xz/yz electrons forming the upper and lower Hubbard band as seen by ARPES and DMFT. In addition, the interpretation of XAS is complicated by the large rotations of the octahedra leading to a deviation of the local coordinates, in which we define the orbital character, and the global crystalline axes. These rotations, together with the strong effects of spin-orbit coupling (SOC) on the Fermi surface are also likely the reason why our linear dichroism experiments in ARPES were not fully conclusive.

This leaves a comparison of the experimental Fermi surface with DMFT as the most promising route to obtain precise orbital occupations. In the original manuscript we did not include such a comparison because the agreement with our first set of DMFT calculations that did not include spin-orbit coupling (SOC) was not fully satisfactory. Motivated by the comments of the referees, we have in the meantime completed an extensive set of additional DMFT calculations which use the methods described in Kim et al. PRL 120, 126401 (2018) to include SOC. The Fermi surface obtained from these calculations has been included in the revised manuscript and is in good agreement with the experimental data, which implies that the DMFT occupation numbers in the metallic phase provide an accurate picture. The new calculations thus confirm the abrupt change of orbital occupations at the MIT. In addition, they demonstrate that our ARPES experiments are largely representative of the bulk electronic structure.

To include these new calculations in the manuscript, we split Fig. 2 into two smaller figures. We also added a description of the technical aspects of the calculations under ‘Methods’.

Minor comments:

1. My understanding is that they used Pr doped system which is closer to the MIT and therefore easier to move across the phase boundary by strain. If it is the case, it needs to be explained in the introduction.

This is indeed correct and we have added additional text to the second paragraph of the introduction to make this point clearer.

2. In the 2nd paragraph, the authors state “Consistent with a previous study on La-doped Ca₂RuO₄ [28].....” In that statement, they claim that La and Pr do not introduce itinerant carriers. However, A site rare-earth atom substitution introduces carriers in similar systems such as (La-Sr)₂RuO₄. In addition, in the work in reference 28 only asserts that the doped electrons do not contribute to the magnetism but does not rule out possibility for carrier doping. So, either the claim should be backed by evidences or the statement needs to be revised?

We consider the highly insulating nature of Pr-doped Ca₂RuO₄ shown in Fig. 1b (and fully consistent with Ref. 28 cited by the referee) to be the best evidence for localized dopant electrons. As shown in the figure below, the resistivity of Pr-doped Ca₂RuO₄ differs fundamentally from lightly doped cuprates or iridates, where doped electrons or holes rapidly delocalize. At low temperature, doped Ca₂RuO₄ is at least 5 orders of magnitude less conductive than cuprates or iridates with comparable doping, giving strong evidence for a fully localized character of the additional doped electrons. In the revised version, we include the below figure in supplementary information.

Figure 1: Comparison of the resistivity of lightly rare earth doped Ca₂RuO₄ with comparable doping levels in cuprates and iridates.

A localized nature of the dopants is also fully consistent with our ARPES experiments shown in Fig. 3b (now Fig. 4b), which do not show any evidence for itinerant carriers at doping levels where iridates or cuprates show clear signatures of mobile carriers in the low-energy excitations.

3. The authors say “However, it appears plausible to interpret the extended straight sections of the experimental Fermi surface as originating predominantly from the quasi-1d xz,yz orbitals while the curved sections of the lens-pockets as well as the circular pocket at Gamma are likely of dominant xy character.” It may not be easy for the readers to understand all these if they are not familiar with the electronic structures of ruthenates. For example, it would be easier to understand if lens-pocket and quasi-1d band are marked and described.

We fully agree with this comment and have updated the relevant figure panel following the suggestions by the referee.

4. It is somewhat peculiar to see the Fermi surface in Fig 2b FS indicates very small hybridization between 'yz' and 'xz' bands in spite of the rotation. Is it consistent with DMFT? Can surface be different from bulk?

As shown in the new Fig. 2c, the experimental FS is in good agreement with DMFT that includes spin-orbit coupling. This also suggests that our ARPES experiments are largely representative of the bulk.

5. Technical question 1: Not all the strain in CuBe shown in Fig 1d may be transferred to the sample because epoxy can have elasticity? XRD has been measured to obtain the true strain but one should not assume the strains on the surfaces of CuBe and sample are the same.

We agree that strain relaxation in the epoxy is a serious issue in (almost any) strain dependent experiment. In our experiments this problem is minimized since straining the substrate (as shown in Fig. 1d) relaxes the initial compressive strain in the sample, which we measured directly. Hence, the forces on the epoxy *decrease* while bending the substrate. It thus appears plausible to neglect changes in the strain relaxation during the bending experiment.

6. Is sample bent at all? Is bending truly negligible?

Bending the substrate does bend the sample. However, the minimal bending radius of approximately 25 mm is large compared to the sample dimensions. Over the 50 micron spot probed in our experiment the induced sample curvature translates into a variation of local crystallographic axes by approximately 0.1° , which is negligible.

Reviewer 2

This paper mainly reports the evolution of the electronic states across the MIT in $\text{Ca}_{2-x}\text{Pr}_x\text{RuO}_4$ based on ARPES measurements. There are three main tuning parameters used in this study: {1} Light electron doping with Pr^{3+} in place of Ca^{2+} ; {2} (nearly) biaxial substrate-strain with the BeCu substrate; and {3} (additional) uniaxial bending-strain with a new device. The effects of these three parameters are presented in a mixed and confusing way, as I describe below. Especially one of the main conclusions that the observed metallic state using {1}, {2} and {3} "represents the intrinsic metallic phase of Ca_2RuO_4 " (the last paragraph on page 7) is questionable. Thus, although the paper contains interesting new results, I do not recommend it for publication in Nature Communication, unless major revision is made to the manuscript. Below, I list specific comments for revision.

The concerns of referee 2 about the presentation of our strain experiments are largely based on a misunderstanding of a crucial detail of our study.

The strain in our experiment has two contributions i) different thermal expansion of substrate and sample (ϵ^i in our notation) and ii) strain induced by bending the substrate (ϵ^{sub}). While it appears natural to assume that the former is biaxial, as done by the referee, this is clearly not correct as can be seen from the X-ray data in our Fig. 1f. Because of the extremely anisotropic thermal expansion of Ca_2RuO_4 , the strain ϵ^i induced by cooling through the phase transition is to a good approximation uniaxial.

In fact, its calculation is straightforward from the crystallographic data. Along the b-axis we find a contribution due to thermal expansion of the sample of $\epsilon_{xx}^i = (b^{300K} - b^{10K}) / b^{10K} = -3.8\%$ for $x=0.04$, which was already stated in the supplementary. Using the same method we find $\epsilon_{yy}^i = (a^{300K} - a^{10K}) / a^{10K} = -0.19\%$ for the initial strain in the orthogonal in-plane direction. Adding the contraction of the substrate of $\Delta L/L \approx -0.28\%$, we obtain $\epsilon_{xx}^i \approx -4.1\%$ and $\epsilon_{yy}^i \approx -0.47\%$. Given that these values differ by approximately an order of magnitude, we believe it is appropriate to call ϵ^i uniaxial.

From supplementary Fig. 2, it can further be deduced that the strain induced by bending the substrate is even closer to being uniaxial over the typical lateral dimensions of our samples of 0.3×0.3 mm. It is thus a good approximation to treat the total strain as uniaxial and write it as the sum of these two contributions.

In the revised manuscript, we added the calculation of ϵ_{yy}^i to the supplementary, to provide quantitative information on the anisotropy of the in-plane strain. In addition, we changed our notation and now use ϵ^{bend} instead of ϵ^{sub} as recommended by the referee under point (5).

To clarify the role of Pr substitution {1}, we have added additional text in the introduction, to explain more clearly that the main effect of Pr is to move the system closer to the MIT. We now also discuss the localization of the extra Pr electrons in the Mott-insulating S-Pbca phase in more detail in the main text and supplementary.

We hope that these comments also answer points (3), (4), (5) and (11).

The second main point raised by referee 2 concerning the “intrinsic metallic phase” of Ca_2RuO_4 is addressed below under point (7).

(1) In the title, “Ca₂RuO₄” should be changed to “Ca₂-xPr_xRuO₄”, since the main results in this paper is not based on “pure Ca₂RuO₄”.

We are open to such a change but would like to leave this decision to the editor. Clearly the suggestion by the referee is more precise. On the other hand, our title is more readable and, we believe, catches the essence of our study.

(2) In the abstract, “previously unexplored metallic state at low temperatures” should be rephrased to “previously unexplored electronic structures of the metallic state at low temperatures”, since there are already a few reports on pressure-induced metallic state, ref (23) and Alireza et al., J. Phys.: Condens. Matter 22 (2010) 052202. The latter reported superconductivity as well.

We agree with the referee and have changed the abstract accordingly.

(3) The abstract should clearly state the roles of {1}, {2} and {3} above. The present abstract states as if “uniaxial strain” (the first sentence) of -4.1% has been achieved. In my understanding, -4.1% refers to biaxial substrate-strain and also in the ab-plane (not mentioned).

See the above reply to the main comment of referee 2.

(4) In captions of Fig. 1, again the roles of {1}, {2} and {3} are mixed and confusing. Figs. a and b represent {1}. Figs. e and f represent a “standard substrate effect” due to {2} without “bending”. Figs. c and d are related to “additional” strain by {3}. Thus, it is natural to align (a,b)(f,e)(c,d) in this order.

See the above reply to the main comment of referee 2.

The explanation of the inset of Fig. e is needed in the caption.

The inset of Fig. 1e shows a close up of the marked area. We now state this explicitly in the caption.

The notation ϵ^{sub} in Fig. d is confusing. See (5) below.

See the above reply to the main comment of referee 2.

(5) On page 5, the major strain introduced in this study, ϵ^{I} , is biaxial substrate-strain due to differential thermal contraction between the sample and the BeCu substrate. According to Fig. 1 f, the a-axis is also strongly compressed. Thus I suggest using $\epsilon^{\text{total}} = \epsilon^{\text{substrate}} + \epsilon^{\text{bending}}$ or $\epsilon^{\text{total}} = \epsilon^{\text{biaxial}} + \epsilon^{\text{uniaxial}}$ to clarify the notations. By the way is it correct to write $\epsilon^{\text{i}_{xx}} = \epsilon^{\text{i}_{yy}} = -4.1\%$ here? If the assessment of $\epsilon^{\text{i}_{yy}}$ is difficult, the author should write so in the main text; I notice no description in the Supplement, either.

See the above reply to the main comment of referee 2.

(6) Concerning the first line on page 5, a very relevant literature explaining the anisotropic effects of uniaxial in-plane strains on T_N is Taniguchi et al., Phys. Rev. B 88, 205111 (2013), "Anisotropic uniaxial pressure response of the Mott insulator Ca_2RuO_4 ".

We agree with the referee that it is appropriate to cite this paper and have done so in the revised manuscript.

(7) The last paragraph on page 7: Persuasive argument is need to explain why the doped and uniaxially strained system "represents the intrinsic metallic phase of Ca_2RuO_4 ", rather than hydrostatically compressed pure Ca_2RuO_4 with the ferromagnetic metallic state and superconductivity.

We chose this formulation in the original manuscript because the strained state of Pr-doped Ca_2RuO_4 is paramagnetic just like the high-temperature metallic phase of pure and doped Ca_2RuO_4 . However, the referee's comment made us realize that this argument is not strong. Naturally, a paramagnetic high-temperature state does not exclude a ferromagnetic ground state. In addition, ferromagnetism has also been observed in pure Ca_2RuO_4 under uniaxial pressure making the observation in our work of a low-temperature paramagnetic phase even more remarkable. In view of these points, we have rephrased the statement criticized by the referee and improved the discussion of previous thermodynamic and transport measurements of Ca_2RuO_4 under pressure.

(8) The first paragraph on page 8: The discussion and conclusion in this paper heavily rely on the assignment that "the curved sections of the lens-pocket ... are likely of dominant xy character". If the authors mean that the other half (straight sections) of the lens-pocket is of yz/zx character, they should write it clearly. Using two different colors in the extracted Fig. 2 c may help.

As described in our reply to referee 1, we have strengthened the conclusions based on the assignment of orbital characters of Fermi surface sheets by adding new DMFT calculations that include spin-orbit coupling. We also changed the color scheme of Fig. 2c as recommended by the referee.

(9) In the third paragraph on page 8: "The first important conclusion from this data is that light Pr-doping alone causes minor changes in the electronic structure only." This is clearly an under-evaluation of the doping effect, since ferromagnetic ordering disappears in the metallic state under strains.

We would like to point out that this sentence refers to the ground state of doped samples, which is a Mott insulator with canted antiferromagnetism in pure and lightly Pr doped Ca₂RuO₄. It thus seems appropriate to use the phrase "minor changes only".

(10) Comments to improve the figures and captions:

10-1. The compound name Ca_{2-x}Pr_xRuO₄ should be clearly written in the captions of Figs. 2 a,b, and Fig. 3.

We now repeat the compound name in addition to the doping values that were already given in these captions.

10-2. Some symbols are not visible due to poor color combinations: The use of red color in the dark background in Figs. 1 a and inset of e, and the use of black "2" in the dark blue background in Fig. 2 d.

We have changed these colors as requested by the referee.

10-3. Fig. 2 d: Is "epsilon" here epsilon^{total}? "%" should be deleted. Since warming severely reduces the magnitude of epsilon^l, the straight line 2-3 should be curved towards positive epsilon on increasing T. More importantly, do the author anticipate paramagnetic insulator state persisting to T=0 as Fig. 2 d indicates? Is it based on the J_{eff}=0 model [18-21]? If so, are the DMFT results in Fig. 3 d and e consistent with the J_{eff}=0 model (in terms of spin-orbit interaction and crystal-field splitting within t_{2g}) ?

We thank the referee for bringing our attention to the schematic phase diagram of Fig. 2d. It was indeed unfortunate to suggest the possibility of a paramagnetic insulating ground state in this sketch. A complete absence of magnetic order in the insulating phase is exceedingly unlikely and our data do not show any evidence for such a spin-liquid phase. We have thus modified the diagram accordingly.

Regarding the arrow 2 -> 3 in the schematic phase diagram: Throughout the paper we define strain values relative to the S-Pbca ground state. With this definition, which we believe to be the cleanest and most intuitive, the path 2 -> 3 does follow a straight line as indicated in our sketch.

(11) The roles of {2} and {3} are not clearly stated and sometimes misleadingly used.

11-1. Fig. 2 g caption: epsilonⁱ ~ - 0.6% here may be epsilon^{total}.

11-2. Second paragraph on page 9: "Straining" here means biaxial substrate-strain since Fig. 3 are all about {2} WITHOUT {3}.

As argued above, {2} and {3} are both uniaxial to a good approximation, which justifies our use of $\epsilon^{\text{tot}} = \epsilon^i + \epsilon^{\text{sub}}$. (ϵ^{bend} . in the revised version).

(12) In Fig. 1 b for x (Pr) = 0.07, why does rho first drops just below 100 K?

We attribute this to a micro-crack appearing near the phase transition. Since this drop is not essential for the conclusions of the paper we have not repeated the resistivity measurements to eliminate this minor artifact.

Reviewer 3

We thank the reviewer for the overall very positive comments and hope that he/she appreciates the additional calculations in the revised manuscript that include correlation enhanced spin-orbit coupling and go substantially beyond published DMFT results. Importantly, these calculations provide for the first time a good quantitative description of the fermiology of metallic Ca_2RuO_4 , which allows us to draw much firmer conclusions regarding the orbital polarization in this system.

REVIEWERS' COMMENTS:

Reviewer #1 (Remarks to the Author):

Ricco et al have resubmitted the manuscript. My biggest concern was about the actual occupation of each orbital in the insulating states. The authors have made plausible argument about (1) why XAS may not be the real indicator of the orbital occupation (such as octahedron distortion), and (2) why DMFT results may be used. They revised manuscript accordingly (Fig 2 => Figs 2 & 3).

There were some confusing points about the notations for strains as pointed out by the referee 2, which has been addressed by the authors.

Only 1 minor comment for the authors to think about:

Regarding comment #4 that I raised in the first report, I also would say that the agreement between DMFT and exp is fairly good. However, the hybridization between xz & yz bands is much smaller in the experiment than in theory, judging from the FS topology. Please note that the two FSs (large and small square-type) are almost touching each other in the experimental data but are completely detached in the theory on the Γ -X/Y line. The detachment comes from $xz - yz$ hybridization but it looks very small for (strained) experimental data.

Reviewer #2 (Remarks to the Author):

The revised version is substantially improved. I recommend publication if the following is well considered.

(1) There is no mention of Pr-doping in the title and abstract.

I strongly suggest at least to include this information on the 2nd line in the abstract.

"lightly doped bulk Ca_2RuO_4 with uniaxial strain."

>>> "bulk Ca_2RuO_4 lightly doped with Pr under uniaxial strain."

(3) I now understand that the claimed -4% strain is simply by suppressing the unusual negative thermal expansion of the sample by substrate. Additional statements made on page 3 help to avoid misunderstanding. To further clarify the experimental situation, I suggest adding "uniaxial" on the 4th line in the abstract,

"we achieve strain" >> "we achieve uniaxial strain".

(10) Comments to improve the figures and captions:

> 10-1. The compound name $\text{Ca}_{2-x}\text{Pr}_x\text{RuO}_4$ should be clearly written in the captions of Figs. 2 a,b, and Fig. 3.

In the caption of new Fig. 3, the name of the compound is still missing. Are the data all from $\text{Ca}_{2-x}\text{Pr}_x\text{RuO}_4$ with $x=0.07$?

>11-1. Fig. 2 g caption: $\epsilon^i \sim -0.6\%$ here may be ϵ^{total} .

In the caption of new Fig. 3, is it really $\epsilon^i \sim -0.6\%$, instead of $\epsilon^{\text{total}} \sim -0.6\%$? Is there no additional bending here?

Additional minor comment: There are many errors in the use of Italic and Roman characters.

"mott" in Refs. 30 and 35 should be "Mott".

End

Reviewer #3 (Remarks to the Author):

In their revised manuscript, Ricco et al. elaborated on the points of criticism of the referees. However for some reason, they did not truly respond to my (referee 3) critical comments and suggestions, which does not look very respectful.

Furthermore, the additional incorporation of spin-orbit coupling (within a certain approximation) in the calculations is to be appreciated, but it is questionable if it is truly needed for the here discussed physics. In other words, clarification of the orbital occupations via an improved fermiology appears as a somewhat odd angle. In a recent work by Zhang and Pavarini (PRB 95, 075145 (2017)), it was claimed that spin-orbit coupling is not decisive for a metal-insulator transition in Ca_2RuO_4 . Instead, a more thorough theoretical modelling of the induced strain on the first-principles level, and a monitoring of the orbital occupations along this route, was suggested in the referee report.

But the authors do not comment at all on this. Then, there were questions on the local J_{eff} states with spin-orbit coupling by the other referees, which are also not really addressed, even if that coupling is now included.

There is the slight impression, that the authors decided to do 'something' on the theory side and chose the most convenient option, since if one tracks the pure-theory work of the related authors, spin-orbit coupling is a close research interest.

So from my side, am back at the original picturing of the work from the first report: On an experimental/technical level it is surely of highlighting character, but the physics outcome, especially also in view of a better understanding of the change of orbital occupations with strain in this compound, is limited. This is not to say that this work is physics-wise not interesting, but there are high standards for Nature Communications, which are not truly met from that perspective.

Below, we reply to all criticism raised by the referees. We copy the original referee reports in black and provide a response in red.

REVIEWERS' COMMENTS:

Reviewer #1 (Remarks to the Author):

Ricco et al have resubmitted the manuscript. My biggest concern was about the actual occupation of each orbital in the insulating states. The authors have made plausible argument about (1) why XAS may not be the real indicator of the orbital occupation (such as octahedron distortion), and (2) why DMFT results may be used. They revised manuscript accordingly (Fig 2 => Figs 2 & 3).

There were some confusing points about the notations for strains as pointed out by the referee 2, which has been addressed by the authors.

Only 1 minor comment for the authors to think about:

Regarding comment #4 that I raised in the first report, I also would say that the agreement between DMFT and exp is fairly good. However, the hybridization between xz & yz bands is much smaller in the experiment than in theory, judging from the FS topology. Please note that the two FSs (large and small square-type) are almost touching each other in the experimental data but are completely detached in the theory on the Gamma-X/Y line. The detachment comes from xz - yz hybridization but it looks very small for (strained) experimental data.

We agree with the referee that this particular non-crossing gap is underestimated in our calculations. However, this discrepancy has a negligible effect on the orbital polarization and on any other statements made in our manuscript. Moreover, its origin is complex and not fully understood. xz/yz hybridization, as pointed out by the referee is important but is not the only effect. In addition, the gap is affected by spin-orbit coupling and by the admixture of xy orbital character. We believe that discussing the limitations of the theoretical treatment of all these factors is overly specialized and have thus refrained from doing so in the manuscript.

Reviewer #2 (Remarks to the Author):

The revised version is substantially improved. I recommend publication if the following is well considered.

(1) There is no mention of Pr-doping in the title and abstract.

I strongly suggest at least to include this information on the 2nd line in the abstract. "lightly doped bulk Ca_2RuO_4 with uniaxial strain."

>>> "bulk Ca_2RuO_4 lightly doped with Pr under uniaxial strain."

We have followed the advice of the referee and included this information.

(3) I now understand that the claimed -4% strain is simply by suppressing the unusual negative thermal expansion of the sample by substrate. Additional statements made on page 3 help to avoid misunderstanding. To further clarify the experimental situation, I suggest adding “uniaxial” on the 4th line in the abstract, “we achieve strain” >> “we achieve uniaxial strain”.

We specified that the strain is uniaxial.

(10) Comments to improve the figures and captions:

> 10-1. The compound name $\text{Ca}_{2-x}\text{Pr}_x\text{RuO}_4$ should be clearly written in the captions of Figs. 2 a,b, and Fig. 3.

We specified the name of the compounds in the captions of Fig. 2 and 3.

In the caption of new Fig. 3, the name of the compound is still missing. Are the data all from $\text{Ca}_{2-x}\text{Pr}_x\text{RuO}_4$ with $x=0.07$?

We repeated the name of the compound. All data in this figure are from $x=0.07$.

>11-1. Fig. 2 g caption: $\epsilon^i \sim -0.6\%$ here may be ϵ^{total} .

We thank the referee for pointing out this mistake. We have changed the caption accordingly.

In the caption of new Fig. 3, is it really $\epsilon^i \sim -0.6\%$, instead of $\epsilon^{\text{total}} \sim -0.6\%$? Is there no additional bending here?

The referee is correct. This should be the total strain. We changed the labels in the figure accordingly.

Additional minor comment: There are many errors in the use of Italic and Roman characters. “mott” in Refs. 30 and 35 should be “Mott”.

We corrected the font style and the mistake in the references.

End

Reviewer #3 (Remarks to the Author):

In their revised manuscript, Ricco et al. elaborated on the points of criticism of the referees. However for some reason, they did not truly respond to my (referee 3) critical comments and suggestions, which does not look very respectful. Furthermore, the additional incorporation of spin-orbit coupling (within a certain approximation) in the calculations is to be appreciated, but it is questionable of it is truly needed for the here discussed physics. In other words, clarification of the

orbital occupations via an improved fermiology appears as a somewhat odd angle. In a recent work by Zhang and Pavarini (PRB 95, 075145 (2017)), it was claimed that spin-orbit coupling is not decisive for a metal-insulator transition in Ca_2RuO_4 . Instead, a more thorough theoretical modelling of the induced strain on the first-principles level, and a monitoring of the orbital occupations along this route, was suggested in the referee report. But the authors do not comment at all on this. Then, there were questions on the local J_{eff} states with spin-orbit coupling by the other referees, which are also not really addressed, even if that coupling is now included.

There is the slight impression, that the authors decided to do 'something' on the theory side and chose the most convenient option, since if one tracks the pure-theory work of the related authors, spin-orbit coupling is a close research interest. So from my side, am back at the original picturing of the work from the first report: On an experimental/technical level it is surely of highlighting character, but the physics outcome, especially also in view of a better understanding of the change of orbital occupations with strain in this compound, is limited. This is not to say that this work is physics-wise not interesting, but there are high standards for Nature Communications, which are not truly met from that perspective.

We thank the referee for their insightful comments. Both, the inclusion of strain in the theoretical description as well as the issue of $J_{\text{eff}}=0$ versus $S=1$ states currently being discussed in the literature are undoubtedly interesting. However, the latter is not within the topic of our work, as we do not probe magnetism directly. The former is closely related to our results but is a fundamental theoretical challenge beyond the scope of our mainly experimental work. Moreover, existing theoretical approaches cannot treat phase separation as it is observed in our experiments, which would reduce the value of a full theoretical study in this particular case.